# Novel Composites of Poly(vinyl chloride) with Carbon Fibre/Carbon Nanotube Hybrid Filler

**DOI:** 10.3390/ma15165625

**Published:** 2022-08-16

**Authors:** Katarzyna Skórczewska, Krzysztof Lewandowski, Sławomir Wilczewski

**Affiliations:** Faculty of Chemical Technology and Engineering, Bydgoszcz University of Science and Technology, Seminaryjna 3, 85-326 Bydgoszcz, Poland

**Keywords:** poly(vinyl chloride), carbon fibre, hybrid filler, polymer matrix composites (PMCs), properties

## Abstract

This article presents the results of studies of poly(vinyl chloride) (PVC) composites modified with a hybrid carbon filler of carbon fibres (CFs) and multiwalled carbon nanotubes (MWCNTs). The hybrid filler was produced by a solvent method, using poly(vinyl acetate) (PVAc) as an adhesive. The proportion of components in the hybrid filler with CF–CNT–PVAc was 50:2.5:1, respectively. The obtained hybrid filler was evaluated by SEM, TG, and Raman spectroscopy. The PVC composites were produced by extrusion with proportions of the hybrid filler as 1 wt%, 5 wt%, or 10 wt%. Thermal stability by the TG method, mechanical properties, and the glass transition temperature (T_g_) by the DMA and DSC methods were determined. The composite structure was evaluated by SEM and Raman spectroscopy. The effect of the hybrid filler on electrical properties was investigated by studying the cross and surface resistivity. It was concluded that, aside from a substantial increase in the elastic modulus, no substantial improvement in the PVC/CF/CNT composites’ mechanical properties was observed; however, slight increases in thermal stability and Tg were noted. The addition of the hybrid filler contributed to a substantial change in the composites’ electrical properties. SEM observations demonstrated improved CNT dispersibility in the matrix, however, without a completely homogeneous coverage of CF by CNT.

## 1. Introduction

Poly(vinyl chloride) is a polymer characterised by high chemical resistance and resistance to weather conditions [1,2]. This polymer can undergo many physical modifications without using complex methods, thereby adapting it to specific requirements related to its intended purpose [3,4]. In 2021, the global poly(vinyl chloride) (PVC) capacity was 60.27 MPTA [5], and the demand for this raw material is constantly increasing. PVC is characterised by high chemical resistance, favourable mechanical properties, as well as resistance to water and weather conditions and good adhesive properties [4,6,7]. For economic reasons and due to its favourable performance characteristics, the material is commonly used in many branches of the economy, especially in construction [8,9,10,11,12], the electrical industry [13,14,15], and in food packaging [16]. One of the main inconveniences of using PVC is its weak thermal stability; therefore, the polymer requires using thermal stabilisers [17,18].

Poly(vinyl chloride) is a material often tested as a matrix for composites modified using natural fillers [19,20,21,22] or modern polymer nanocomposites intended for special applications [23,24,25,26,27]. Many papers describe PVC composites modified with natural fibres, such as: sisal fibres [28], coconut fibres [29], wood fibres [30], or mineral fibres, such as glass fibres [31,32]. However, there is little information on PVC composites with carbon fibre [33,34,35,36,37,38,39].

The use of nanocarbon materials in polymer modification and the assessment of such materials’ properties constitute a common research topic [40,41,42,43]. By using such nanoadditives, it is possible obtain materials with improved features suitable for new applications even at small shares. The impact of graphene [24,44], fullerenes [45], and carbon nanotubes [45,46,47] on PVC properties has been discussed in many papers. When analysing the literature coverage, it is possible to note that the key issue with nanocomposite production is achieving a homogeneous nanofiller dispersion in the polymer matrix, which plays a fundamental role for new polymer nanocomposite properties and impacts the nanofiller–polymer macromolecule interface. Much research is being devoted to improving the nanofiller’s dispersibility by using methods of physical and chemical modification.

Carbon nanotubes, compared to steel, show 50 to 100 times higher tensile strength. Compared to carbon fibres, nanotubes exhibit higher stiffness and lower brittleness. The elasticity of the graphene layer makes nanotubes extremely flexible. The electrical properties of nanotubes are mainly influenced by their size and chirality. They conduct electricity and heat very well along the axis, but poorly in the transverse direction. CNTs, due to their polycyclic structure, also exhibit low chemical reactivity [48,49].

Carbon fibres (CF) are used in various forms as polymer material reinforcement [50,51,52]. Due to their mechanical and thermal properties as well as a favourable strength–density ratio, these composites (carbon fibre reinforced polymers(CRFP)) are used in the aviation, automotive, sport, bio-medical, and electronic industries. Due to their highly crystalline graphite surfaces with inert structures, [53] carbon fibres are characterised by a non-polar, inert, and relatively smooth surface. Carbon fibres demonstrate weak wettability toward polymers [51], potentially leading to delamination and low mechanical strength of composites if suitable steps are not taken [54]. Many techniques of improving CF wettability have been developed. One possibility of achieving suitable adhesion at the interface, enabling the transfer of loads in a polymer–fibre interface, is to apply sizing to fibres [55], including applying carbon coatings in the form of carbon nanotubes [56,57,58,59,60] or graphene [58,61,62].

One such method involves using carbon nanostructures by coating CF surfaces with CNTs to ensure higher interface adhesion between the CF and the polymer matrix. The CNTs can be applied to the CF by chemical vapour deposition (CVD), where the CF acts as the base on which the CNTs are created from a volatile precursor; electrophoretic deposition (EPD), which deposits charged molecules dispersed in a liquid suspension under the applied electrical field; electrostatic spray deposition (ESD), where aerosol spraying using an electric field is aimed at the target location; and chemical functionalisation or physical modification [54]. Chemical functionalisation relies on the presence of functional groups on carbon structure surfaces and their reactivity. The selection of functional groups in CF and CNT is deemed as the most important factor in CF–CNT hybrid filler production. Chemical functionalisation most often occurs at locations of structural defects or in oxidised structures. Strong sulphuric, nitric, or hydrochloric acids, mixtures containing potassium permanganate, ozone or hydrogen peroxide are used for oxidation purposes [63,64]. Deposited groups constitute the basis for further reactions, e.g., silanating. These processes simultaneously cause structure defects and can affect the properties of carbon materials and composites containing carbon materials, especially in the case of electrical properties [65]. Therefore, in order to retain the unique properties of nanostructures, it is possible to use non-covalent functionalisation techniques, i.e., through physical impacts, e.g., by using dispersers or coating-forming materials/other polymers with the use of immersion or spraying methods [66,67]. These methods are most often used to modify CF in the form of mats and textiles and are less often used to modify cut fibres. 

The use of CNTs in composites containing carbon fibres improves the material’s strength, thermal stability, and mechanical properties [56,57,58,59,68,69,70,71,72,73]. In addition, this hybrid micro-carbon–nanofiller interface is favourable to a more effective nanofiller dispersion in the matrix. This type of hybrid composite was obtained by the simultaneous introduction of both fillers to a molten polymer [74,75,76] or the introduction of carbon nanotubes to a polymer matrix prior to adding carbon fibres [77]. In other works, nanoadditives were placed on carbon fibres before polymer composites were produced [68]. 

Some papers state that nanoadditives can improve the adherence of fibres to polymers, thereby ensuring more favourable mechanical properties of composite coatings [73]. In another paper, it was suggested that the use of a CF–CNT hybrid interface contributes to reductions in CNT aggregate quantities observed in PA nanocomposites obtained by mixing plasticised components when compared to PA/CNT nanocomposites. At the same time, the filler–matrix interface surface was greater than in the nanocomposites [75,78]. 

Despite the fact that papers describing PVC composites with carbon fibres are known [33,34,35,36,37,38,39], the literature does not feature coverage of using a carbon fibre–carbon nanotube hybrid filler to modify PVC. The use of the CF/CNT hybrid filler can lead to the development of advanced polymer materials with perfect parameters. Combining carbon fibres with carbon nanotubes can lead to changes in the mechanical and electrical properties and chemical resistance of the obtained composites [66,79,80].

This paper describes the testing of a PVC composites modified with a CF/CNT hybrid filler. The hybrid filler was produced by physical modification between the carbon fibre and non-oxidised carbon nanotubes with the use of the PVAc adhesive compound. The noted trends based on the Scopus database demonstrate that studies on CF–CNT hybrid fillers are gaining in popularity. 

## 2. Materials and Methods

### 2.1. Materials

A dry blend of poly(vinyl chloride) (PVC) with the composition of PVC S61 (Anwil SA, Włocławek, Poland) (100 phr), thermal stabiliser Mark MOK 17M (Acros, Renningen, Germany) (4 phr), and wax Loxiol G22 (Henkel, Dusseldorf, Germany) (1 phr) was used as the matrix for the investigated composites.

Multiwalled carbon nanotubes NC7000 (MWCNT) (Nanocyl S.A., Sambreville, Belgium) of industrial grade with an average diameter of about 9.5 nm, an average length of 1.5 microns, and a maximum length of 50 μm were used to obtain a hybrid filler for PVC modification. 

The hybrid filler was produced with the use of the commercial milled carbon fibres (CF) ZOLTEK PX 35, (ZOLTEK™ Toray Group, Bridgeton, MO, USA) with an average diameter of 7.2 μm and an average fibre length of 150 μm. 

Poly(vinyl acetate) (PVAc) with a molecular weight (Mw) of approx. 50.000 (Alfa Aesar, Thermo Fisher Scientific, Kandel, Germany) was used as the adhesive agent in the hybrid filler’s modification. PVAc is commonly used as a thermoplastic adhesive. Its purpose is to modify the surfaces of carbon fibres [78]. PVC/PVAc mixtures and copolymers are also known [81]. This polymer demonstrates good CNT dispersion properties [82].

Acetone with a purity of 99.5% (Pol-Aura, Dywity, Poland) was used as the solvent and environment for obtaining the hybrid filler.

### 2.2. Production of the Hybrid Filler

The first stage involved the production of a carbon hybrid filler. In order to improve the dispersion’s homogeneity, the carbon nanotubes underwent sonication in acetone for 10 min at a temperature of 20 °C by ultrasound with a frequency of 20 kHz and an amplitude of 40%, using a SONOPULS homogeniser from Bandelin, equipped with a rod probe. Then, the homogeneous dispersion was introduced into the carbon fibres in acetone, and the entire solution was mixed with a mechanical mixer (Ika Eurostar 6000) for 10 min. Then, the PVAc solution was introduced, and the mixing continued for another 10 min. The obtained mixture was transferred to a glass tray and was dried at 60 °C for 24 h in a drier with forced air flow and then in a vacuum drier at 60 °C for another 24 h. This allowed for obtaining a micro-nanocarbon hybrid filler in which the ratio of carbon nanotubes, carbon fibres, and PVAc was equal to 2.5:50:1.

### 2.3. Production of PVC Composites with the Hybrid Filler

PVC composites with the hybrid filler were produced by extrusion and pressing. The first stage involved obtaining dry blend PVC with process additives by mixing for 15 min in a mixer heated up to 100 °C. Then, the right quantity of PVC dry blend was mixed with the hybrid filler in a mechanical mixer (Ika Eurostar 6000) for 10 min. This allowed for obtaining PVC composites containing 1, 5, and 10% of the carbon hybrid filler’s weight. The obtained mixtures were extruded by using the Brabender laboratory extruder with a screw diameter of d = 15 mm and a length of L/D = 14. The extrusion temperature was as follows: 165 °C in zone I, 180 °C in zone II, and 185 °C at the head. The screw’s rotation speed was equal to 50 rpm, and a nozzle with a diameter of 2 mm and a length of 30 mm was used. The extrudates were granulated, and the obtained granulates were pressed at the maximum pressure of 15 MPa at 190 °C into plates with dimensions of 100 mm × 100 mm × 4 mm and 120 mm × 120 mm× 2 mm, which were used to cut samples for further testing. This allowed for obtaining PVC composites containing 1, 5, and 10 wt% of the carbon hybrid filler. 

### 2.4. Characterisation of the Hybrid Filler and PVC Composites

The hybrid filler’s characterisation was carried out by using Raman spectroscopy and scanning electron microscopy. The quantity of deposited PVAc was also tested using the TG method.

In order to assess the impact of the produced hybrid filler on the properties of PVC composites, the time of thermal stability and decomposition temperature were tested. Tensile properties and impact strength were also assessed. Dynamic mechanical testing was conducted using the dynamical mechanical analysis (DMA) method. The assessment also covered the electrical properties and glass transition temperature by differential scanning calorimetry (DSC) and DMA. The structure of the composites was assessed by SEM and Raman spectroscopy.

#### 2.4.1. Thermogravimetric Analysis

Thermogravimetric tests were carried out using a TG 209 F3 Netzsch Group (Germany, Selb) device in the temperature range of 30–900 °C at a rate of 10 °C·min^−1^ in an inert atmosphere. A sample (approx. 10 mg) was taken from pressed plates prepared for mechanical tests. The temperature of thermal stability was determined as the temperature at which the 5% weight loss of the material was observed (T_5_). The temperature at which a 10% weight loss of the sample occurred was also determined (T_10)._ The temperature at which the decomposition with the highest intensity occurred was determined (T_DTG_) from the DTG curves. Undecomposed residue at 900 °C after combustion (RM) was also established. A TG analysis of the filler used was performed in the same conditions.

#### 2.4.2. Determination of Mechanical Properties

The mechanical properties in static tension were determined in accordance with EN ISO 527 [83,84]. The standardised test specimens (type 5A) were cut using a CNC milling machine from a plate with dimensions of 120 mm × 120 mm × 2 mm. The measurement was carried out with a Zwick/Rolel Z010 testing machine at 23 °C. The test speed was 10 mm/min (for modulus 1 mm/min). The modulus of elasticity, maximum stress, and strain at maximum stress were determined.

The impact strength was tested by using the Charpy method in accordance with EN ISO 179-1 [85]. Rectangular specimens with a length of 80 mm and a cross-sectional dimension at the notch of about 8 mm × 4 mm were used. Specimens with notches (type A) were cut from pressed plates using a CNC milling machine. The tests featured the use of a pendulum with a nominal impact energy of 4 J. The measurement was taken for 5 samples.

#### 2.4.3. Determination of Thermal Stability

The time of thermal stability was determined by using the Congo red method. A material was placed in a glass test tube with an inner diameter of 4.7 mm and a wall thickness of 0.65 mm. A Congo test paper was inserted to a depth of 5 mm in the upper part of the test tube. Next, the tube was placed in an oil bath heated up to 200 °C. The test result enabled the designation of the time (in minutes) when the first visible colour change in the test paper was observed, indicating the decomposition of PVC and the evaporation of hydrogen chloride as a PVC decomposition product.

#### 2.4.4. Dynamical Mechanical Analysis (DMA)

The tests by dynamic mechanical analysis (DMA) were conducted with the use of the DMA Artemis Netzsch device (Selb, Germany) operating in a three-point bending layout (support spacing of 20 mm, sample width of 10 mm, thickness of 1 mm), with a strain of 10 μm, a temperature range of 25–120 °C, and a build-up speed of 2 °C·min^−1^. The strain frequency was equal to 1 Hz. A relatively low strain frequency and amplitude enabled taking measurements in a linear range of viscoelasticity and a low loss modulus [86]. Changes in the storage modulus were designated depending on the temperature. The DMA measurements were also used to analyse the hybrid filler’s impact on the composite matrix’s glass transition. The glass transition temperature was designated based on the changes in the storage modulus (E′) loss angle tangent (tan δ) in a function of temperature. The glass transition range was characterised with a quantity designated as the start of a sudden drop in the value of E′ (start) and the maximum value of tan δ [87].

#### 2.4.5. Determination of Glass Transition Temperature by the DSC Method

The Differential Scanning Calorimetry (DSC) method was used to determine the hybrid filler’s impact on the PVC matrix composites’ glass transition temperature. The measurement was taken with the use of the DSC 204F1 Netzsch device (Selb, Germany). The material sample with an approximate weight of 25 mg was placed in the device’s chamber in a punctured crucible, and the measurement was then taken at the temperature range of 20–140 °C in a nitrogen atmosphere with two heating and cooling cycles. The glass transition temperature was designated from the second heating cycle by analysing the inflection point of the observed baseline change (T_g Infl_) and as half the height of the incremental baseline change (T_g Mid_). The samplE′s measurement was repeated twice.

#### 2.4.6. Scanning Electron Microscopy (SEM)

The structure of the produced hybrid filler and the structure of the composites were assessed using the Tescan MIRA3 scanning electron microscope (Tescan, Brno, Czechia). The materials were assessed at an accelerating voltage of 12 kV. A carbon coating with an approximate thickness of 20 nm was applied to the cryogenic fractures of the composite samples and the filler samples by using the Joel JEE 4B vacuum evaporator. 

#### 2.4.7. Filler and Composite Testing by the Raman Spectroscopy Method

Raman spectroscopy was used to analyse the produced hybrid filler and to assess the composites’ structure. The Raman spectra were recorded in the range of 100–3200 cm^−1^ by using the Renishaw InVia microscope, equipped with a semiconductor laser emitting the wavelength of 488 nm. The power of the laser beam focused on the sample, with a 50× objective lens, was maintained below 0.1 mW. The obtained signals were calibrated before data collection using a crystalline Si sample as an internal standard. The Raman spectra were analysed using the Origin 8.6 PRO software.

#### 2.4.8. Electrical Properties

The electrical properties of PVC composites with the carbon hybrid filler were assessed by testing the surface and volume resistivity. The measurement was conducted by using a measurement system consisting of the 6517A electrometer and the 8009 measurement chamber (Keithley Instrument Inc., Cleveland, OH, USA) in air at 23 °C with a relative humidity of 50% and a voltage of 10 V. Samples with the dimensions of 90 mm × 90 mm × 0.4 mm were used in the testing. 

## 3. Test Results and Discussion

### 3.1. Results of the Hybrid Filler Analysis

Figure 1 presents the TG thermogram for the carbon hybrid filler in a nitrogen atmosphere. A loss of sample weight was observed at the temperature range of 300–340 °C due to the decomposition of the applied PVAc layer [88]. On this basis, it was concluded that the PVAc content in the hybrid filler was approx. 2.1 wt%, which is similar to the value compliant with the filler’s preparation stoichiometry (2.3 wt%). The total sample weight change designated at 900 °C was 4 wt%, which can be related to the decomposition of contaminants generated during CNT synthesis [89] or to the decomposition of the less stable amorphous carbon contained in the hybrid filler [90]. It was concluded that the filler was thermally stable in the PVC processing temperature range, and, more importantly, no residue of the solvent used in the filler’s production process was observed.

The SEM images (Figure 2) demonstrate the structure of the obtained filler at 200×, 10 k× and 50 k× magnification. The produced hybrid filler is characterised by a heterogeneous structure. Nanotube aggregates in the form of wrapped paper-like sheets are distributed between the carbon fibres. Their presence is homogeneous in the filler’s entire volume, thereby confirming the CNTs’ uniform dispersion and their proper homogeneity. The carbon nanotubes were also observed dispersed on the CF surface in the form of adhering aggregate sheets. This dispersion is not homogeneous across the CF entire surface, as there are regions with higher CNT concentrations, which can be related to the different impacts at the CF–PVAc–CNT interface. Therefore, in the future, it is necessary to consider modifying the CF surface structure to increase the impacts with the nanofiller. 

The surfaces of the non-coated CF region show typical grooves and ridges on the fibre axis generated during their production. On the other hand, the dispersion of nanotubes on the fibre’s surface generated a specific topography with an increased surface roughness, which can be favourable for polymer interface impacts and which can improve the fibre’s wettability.

The original CNT agglomerates in the form of larger agglomerates of used Nanocyl NC 7000 with a “combed yarn” structure can have very high cohesive strength and therefore can be difficult to disperse homogeneously [91]. On the other hand, the proposed hybrid filler production procedure allowed for obtaining a filler characterised by smaller nanotube aggregates and their homogeneous dispersion across the entire volume. In addition, it can be assumed that it ensured better dispersion in the polymer matrix. Moreover, regions with higher roughness were observed on the carbon fibres’ surfaces by applying an adhesive layer in the form of PVAc, which can also contribute to improving the adherence of the fibre–matrix interface [51].

### 3.2. Results of PVC/Hybrid Filler Testing

#### 3.2.1. Mechanical Properties

Figure 3 presents the results of testing the following tensile properties depending on the share of the hybrid filler in the matrix: the tensile modulus (*E_t_*), tensile strength (*σ_M_*), and elongation at maximum load (*ε_M_*).

*σ_M_* and *ε_M_* decrease along with increasing shares of the filler above 1 wt%. Initially, with little share of the filler, *σ_M_* and *ε_M_* are comparable to an unmodified PVC. An increase in the hybrid filler share in the matrix deteriorates the mechanical properties. *σ_M_* and *ε_M_* decreased by 17% and 38%, respectively, in relation to PVC at 10 wt%. At the same time, the elastic modulus increased by as much as 44% at 10 wt% filler share when compared to PVC.

The introduction of a carbon hybrid filler did not explicitly improve the composites’ mechanical properties. The observed tensile modulus increase was indicative of influences between the PVC and the filler. However, it must be noted that this value is designated at a low strain value (0.05–0.25%) for polymer materials. At higher strain values, the adhesive forces between the matrix and filler are overcome. This can be the result of both the filler’s uneven adhesion and wettability, but also the result of the presence of carbon aggregates, which act as a discontinuity of the composite structure. As indicated by the filler’s SEM images (Figure 2), not all of the surface of the CFs was coated by the CNT–PVAc adhesive compound. The *σ_M_* decrease could be caused by the anisotropic distribution of relatively short fibres (which was confirmed in the SEM analysis); therefore, many of them, non-oriented in relation to the acting tensile force, were not able to transfer the occurring tensile stress. In addition, the filler substantially limited the ability to orientate PVC macromolecules, which explains the *σ_M_* and *ε_M_* decreases. 

A reference to the above conclusion can also be found in the literature. In the case of modifying a plasticised PVC using a continuous CF, a substantial improvement in mechanical properties, especially in σ*_M_*, was observed, however when analysing PVC/short CF properties, such dependencies were not noted [33]. In [76], the simultaneous use of CF and CNT for PA modification did not cause an improvement in the tensile strength, but there was only an increased E_t_, which was linked to the improved nanotube dispersion in the matrix. On the other hand, a synergy effect of improved mechanical properties due to the use of CF and CNT in PA modification was described in [75], where simultaneous improvements in tensile strength and an increase in E_t_ were achieved, which the authors have suggested to be caused, according to the SEM analysis results, by an increased CNT dispersion homogeneity as a result of using CF.

Figure 4 presents the results of the impact test by using the Charpy V-notch method. It was observed that the addition of a carbon hybrid filler to PVC caused a slight decrease in the composites’ impact strength. The impact strength of an unmodified PVC was approx. 3.4 kJ·m^−2^. Composites containing 1 wt% and 5 wt% of filler were characterised by a similar impact strength of approx. 2.6 kJ·m^−2^. In the case of the maximum filler share, the impact strength decreased to 2.2 kJ·m^−2^.

It is worth noting that the obtained results of the composites’ average impact strength are within the limits of the measurement’s standard deviation. These observations can be explained analogously to the changes in mechanical properties at static tension. The impact strength is affected by the applied method of composite production, causing a random arrangement of carbon fibres, and by the interface adhesion, which does not guarantee maximum stress transfer to the carbon fibres. A similar effect was observed in PVC composites reinforced with post-pyrolytic glass fibres with a carbon deposit [92].

#### 3.2.2. Results of Thermal Stability Testing

The processing of PVC and composites on its matrix requires adequate thermal stability of the material. The time of thermal stability is the time in which a material exposed to a high temperature does not become destroyed. It is an important feature of new composite systems on a PVC matrix and determines their suitability for processing [17,92]. Figure 5 presents the dependency between the tested PVC materials’ thermal stability and the hybrid filler content. 

The addition of a carbon hybrid filler in the amount of 1 wt% causes a slight decrease in thermal stability when compared to an unmodified matrix (by approx. 6%); however, a further increase in filler share does not decrease thermal stability. This effect can be caused by the hybrid filler’s higher heat conduction, and thus by faster heat delivery to the entire volume of the polymer matrix, which is related to faster degradation when compared to an unmodified PVC. The time of the thermal stability of PVC composites and an unmodified matrix was within 31–33 min, and the obtained results indicate that the proposed hybrid filler does not cause a critical decrease in thermal stability and can thereby be used in PVC processing. 

The thermal stability of the PVC/hybrid filler composites and an unmodified matrix was also designated using TG. The results are presented in Figure 6. T_5_, T_10_, T_DTG_, and RM designated for all materials are presented in Table 1. The decomposition of PVC and its composites is similar and takes place in two stages. The first stage of PVC destruction at 250–350 °C is related to the autocatalytic dechlorinating of PVC macromolecules and the simultaneous emission of hydrogen chloride [93]. Additionally, the released HCl by-product additionally catalyses and accelerates the destruction of PVC macromolecules. The next stage, at 350–550 °C, features further decomposition of a secondarily linked polyene structure into non-decomposed carbon residue [92], which, in the case of an unmodified PVC, was approx. 11.3 wt%. In the case of the tested composites, RM is higher due to the presence of a thermally stable filler. 

The analysed materials are characterised by thermal stability exceeding the PVC’s processing temperature of 200 °C, regardless of the filler’s content. The composites are characterised by higher thermal stability defined as the temperature of a 5% sample weight loss (T_5_). For PVC, T_5_ is approx. 254.5 °C, and a filler content of 10 wt% increases this value to 260.1 °C. When analysing T_10_ and T_50_, it is also possible to observe an increasing decomposition temperature along with an increasing share of the filler in the matrix. The composites and the unmodified PVC demonstrate a similar maximum DTG temperature (the maximum difference is 0.8 °C), which indicates the temperature of the most intensive PVC decomposition. It is a very favourable change, as some papers on PVC/CNT nanocomposites describe a decreased thermal stability of composites, caused by the addition of a nanofiller [94,95]. It must, however, be noted that different results of PVC/CNT testing are presented in [96], where the authors argue that the interaction of the isotactic segments of a PVC macromolecule limits the mobility of polymer chains, thereby improving the nanocompositE′s thermal stability. The processing time and temperature undoubtedly affect the thermal stability of PVC and PVC composites.

#### 3.2.3. Results of DMA Analysis

The results of DMA analysis can determine the thermomechanical properties of composites, but they can also specify the modification’s impact on the glass transition and the filler–matrix interface impacts. Figure 7 presents the waveform of changes in the E′ modulus for PVC and PVC/CF/CNT–PVAc composites during bending tests depending on the temperature as well as the waveform of changes in tan δ for the tested materials. Table 2 presents the changes in E′ modulus at 25 °C, 40 °C, 60 °C, 80 °C, and 90 °C as well as the glass transition temperatures. T_g_ is the extrapolated start of a sudden decrease in E′ (T_g E′_) and maximum tan δ (T_g tanδ_). 

The material’s rigidity increases along with increasing content of the carbon hybrid filler in the polymer matrix. In the case of the composite containing 10 wt% filler, E′ at 25 °C increased by 38% when compared to the PVC. The increased rigidity of composites is also retained in the field of the glass transition. In the case of the analysed composite, the E′ modulus at 60 °C and 80 °C is higher by 40% and 98% when compared to the PVC, and the modulus of a sample containing 10 wt% filler at 80 °C corresponds to the PVC’s modulus at 65 °C. At high elasticity following glass transition, the composite’s E′ modulus is over twice as high as in the case of the PVC. These observations indicate an improved elastic behaviour in the viscoelastic field [97,98]. The storage modulus is indicative of the elastic energy stored in the material and is highly affected by its composition, morphology, and geometric characteristics. 

Glass temperature is one of the most important physical and chemical features of amorphous polymers, such as PVC. Glass transition defines a change in the physical and chemical properties of polymers and composites depending on temperature, and it is, therefore, important to define the modifier’s impact on this change [99]. Glass transition is related to the absorption of thermal energy, which makes it possible for segments of polymer macromolecules to shift/move to an extent. Based on the results of the DMA analysis, it was indicated that low contents of the developed filler in the matrix do not cause impact changes at the level of macromolecule segments. A slight increase was, however, observed at the highest filler concentration, which was most likely caused by the presence of carbon nanotubes and CFs affecting the nature of impacts at the molecular level, which is presented in [33,37,88]. The presence of these fillers in the polymer matrix can limit the segmental motion by stiffening the PVC spatial network through deposition in isotactic chain segments [96]. In addition, homogeneously dispersed CNT also reduce the available free space for molecular motion, which is why their production requires more energy. This is observed as an increase in T_g_. The results of the glass transition analysis correspond to the results of the T_g_ tests by DSC.

#### 3.2.4. Results of T_g_ Testing by DSC Method

Figure 8 presents the PVC and composites’ DSC thermograms taken during the 2nd heating cycle and used in the glass transition analysis.

No substantial changes in the tested materials’ T_g Mid_, which changes at 75.7–76.2 °C, were observed (Table 3). The unmodified matrix’s T_g Mid_ was equal to 75.7 °C, and for the composite with the maximum filler share, the value was equal to 76.2 °C, thereby preventing making an explicit conclusion regarding the filler’s impact on the glass transition. On the other hand, an analysis of the glass transition temperature designated as T_g Infl_ allows for the assumption that the filler’s impact on the glass transition is subtle. The values read are within the range of 74.5–77.4 °C. The addition of 10 wt% hybrid filler slightly increases the T_g_ by 2 °C when compared to an unmodified matrix. The observed increase in T_g Infl_ for the PVC/10 CF/CNT-PVAc composite can be caused by the increased share of CNTs introduced along with the hybrid filler and their contribution to the increased T_g,_ which was also described in [87].

#### 3.2.5. Analysis of PVC/Hybrid Filler Structure by SEM Observations

Figure 9 presents examples of SEM images of non-plasticised PVC composites’ cryogenic fractures containing 5 wt% (Figure 9A) and 10 wt% (Figure 9B) hybrid filler and a magnified image of the hybrid filler–polymer matrix interface. The distribution of fibres in the composite is mostly homogeneous, and their arrangement in the matrix does not demonstrate any orientation. The lack of fibre orientation is specific to the applied composite production techniques. It can be noticed that small parts of carbon fibres protrude from the matrix’s surface. The fibres must have, therefore, been broken. This indicates an increased PVC–filler interaction. The different lengths of protruding fibre parts indicate a non-homogeneous adherence between the filler and polymer. 

The magnified image of the fibre–matrix interface (Figure 9C,D) demonstrates the presence of carbon nanotube clusters on the CF surface and along the fibre anchoring in the matrix. The CF–CNT junction at the interface with the polymer matrix demonstrates no clear caverns or cracks, which indicates good wettability when compared to the PVC matrix. Spots with no deposited nanotubes at the CF surface demonstrate clear cracks at the CF/PVC interface.

The SEM image also demonstrates the presence of a fibre–matrix interface, which likely derives from the addition of the PVAc as an adhesive agent (Figure 9D). Its presence is visible in the different interface structure, but also as a rough bladed surface after breaking the modified CF. In addition, the carbon nanotubes deposited on the fibre’s surface increase the fibre’s roughness, thereby improving its adherence to the PVC matrix (better anchoring in the PVC). A similar interface structure appearance was observed in the epoxy composite with CF/CNT [100], where the authors linked the presence of the hackle pattern characteristic of matrix plastic deformation, indicating that the fracture energy was consumed by the plastic deformation of the matrix resin.

#### 3.2.6. Analysis of Electrical Properties

As demonstrated in [23,101,102,103], it is possible to modify electrical properties by using carbon nanofillers. The introduction of carbon fibres also changes the electrical properties of PVC composites, and the conductivity extent depends on the CF’s length [39] and their weight fraction. Moreover, it is not always possible to achieve similar conductivity levels with the same shares of different carbon fibres. In addition, the quantity of nanofillers introduced to achieve conductivity is substantially smaller than in the case of CF. However, a change in the polymers’ electrical properties when using nanotubes requires adequate nanofiller dispersion in the matrix to create conductivity paths across the entire polymer matrix’s volume (percolation network). 

The carbon hybrid filler, containing nano and micrometric particles, can enhance the contact surface of carbon nanotubes deposited on carbon fibres, thereby improve the material’s electron-conducting capacity. Conduction takes place along the generated paths of the conductive filler. The percolation network generation is affected favourably by the filler’s high form factor, which is specific to CNT. An inseparable element of the network generation is the nanofiller’s homogeneous dispersion in the matrix, ensuring higher contact density in the network. In addition, the structure of the obtained hybrid filler demonstrated in the SEM images (Figure 2) highlights CNT dispersion in the form of thin layers/sheets. Such a deposition of the hybrid filler components should enable more effective modification of the PVC composites’ electrical properties.

Figure 10 presents the changes in volume and surface resistivity of composites with a carbon hybrid filler. PVC is a commonly known and used insulator. The non-filled PVC’s volume resistivity (*Rv*) is 1.9 × 10^14^ Ωm. According to expectations, the introduction of 1 wt% hybrid filler lowers resistivity *Rv* to 4 × 10^13^ Ωm. PVC with 10 wt% filler is characterised by lower resistivity Rv by six orders of magnitude. Smaller changes were observed in the case of surface resistivity (*Rs*) due to the hybrid filler addition. The PVC’s surface resistivity was equal to 1.8 × 10^14^ Ω, and the introduction of 5 wt% filler slightly lowered the resistivity to 5.9 × 10^14^ Ω. PVC-10%CF/CNT–PVAc composites are characterised by *Rs* equal to 4.9 × 10^12^ Ω.

Similar results are described in [100], where a small quantity of MWNCT selectively deposited at the CF/epoxy resin interface substantially improved electrical conductivity. The authors suggest, according to SEM observations, that MWCNT’s do not locate flat on the CF surface but are spatially localised in the matrix, which contributes to the formation of spatial conduction paths between the CF and MWCNTs. In the case of hybrid composites of polypropylene with CF and carbon fibres, as presented in [104,105], a substantial increase in conductivity is caused by adding CNT to the matrix. On the other hand, Naji et al. [77] concluded that nanotubes in PVC composites with a carbon microfiller can act as a bridge and increase the distribution density, thereby contributing to the formation of a conduction network. 

The obtained results confirm that a spatial interpenetrating conduction network consisting of micrometric CF and nanometric CNT was formed in the PVC/hybrid filler composites. The CNT dispersed in the form of sheets/bucky paper fragments and CNT dispersion on the CF surface improved the contact density, which ultimately resulted in more effective charge conduction at a smaller nanofiller content according to the conduction mechanism via the CNT structure and direct contact or according to quantum tunnelling. More importantly, the proposed hybrid filler production method without using strong oxidising factors that could actively modify the surface and predispose it to more effective CF–CNT adherence did not breach the carbon nanotubes’ structure, while retaining their electrical conduction potential, which was confirmed by the obtained composites’ lower volume resistivity on the PVC matrix.

#### 3.2.7. Results of Raman Spectroscopy

Raman spectroscopy is used as a tool to determine the structure of carbon materials and to differentiate them in terms of a separate Raman spectral line [106]. In the analysed Raman spectra of CNTs, CF and hybrid filler were recorded at a wavelength of 488 nm, and it was possible to see signals corresponding to bands D and G at 1349–1371 cm^−1^ and 1573–1597 cm^−1^, which were related to unstructured and graphitised carbon, respectively [107]. An analysis of the D and G band (ID/IG) ratio can serve as an indicator of the crystalline order of carbon materials [108] and for evaluating the quality of coverage of CF by CNT [72]. 

The carbon fibres’ Raman spectra (Figure 11) are characterised by broader D and G bands when compared to MWCNT, which is indicative of a lower degree of graphite structure order. The shift in the D and G peaks was caused by fibre grinding and its surface disorder [109]. Aside from the D and G bands, it is possible to see a clear and specific noise 2D band (G’), the presence of which can be related to the CF fibre type with an average Young’s modulus of that of [59]. In the case of the fibres used, the ID/IG ratio was equal to 0.69.

In the case of the CNT spectrum, it is possible to mainly notice band D (≈1350 cm^−1^), band G (≈1580 cm^−1^), and an explicit band 2D (≈2700 cm^−1^). Band D concerns vibration caused by the carbon structurE′s defects. Band G refers to vibration in the perfect carbon structure plain related to sp^2^, and band 2D is the first overtone of band D. The D/G ratio designated for the CNTs used is equal to 0.86. It is indicative of a slight degree of disorder and presence of defects or increased amorphous carbon content, which is specific for technical grade nanotubes obtained by CVD.

The hybrid filler’s Raman spectrum demonstrates the presence of D, G, and 2D peaks. The presence and position of the 2D peak with a Raman shift specific to the CNT used is indicative of the presence of these nanostructures on the carbon fibres [63]. In the case of the hybrid filler, an increased intensity of the G peak and a slight sharpening of the 2D peak were identified, which again confirm the presence of nanotubes on the CF surface. These results correspond with the observations described in [59]. The hybrid filler is characterised by an ID/IG ratio of 0.64. A small change in this value when compared to CF can indicate a physical nature of CF–CNT impact. The hybrid filler production method affected the nanotubes’ dispersion quality. Based on the Raman spectra’s analysis, it can be stated that the nanotubes were deposited on the CF surface.

In the case of composites containing the hybrid filler, bands specific to CNTs, i.e., D and G bands with wavelengths of 1354 and 1587 cm^−1^, respectively, as well as bands specific to PVC, were observed. It is possible to observe a slight shift in the bands specific to PVC, which is indicative of the physical nature of the interactions between the hybrid filler and the matrix’s polymer chains [110]. An increase in the ID/IG ratio intensity (1.07) in the composites when compared to the fillers indicates a good dispersion of the hybrid filler in the PVC matrix.

## 4. Conclusions

The following conclusions were drawn based on the test results:The proposed method of hybrid filler production enabled the deposition of CNTs on the CF surface; however, its distribution was not homogeneous. The CNT aggregates present in the hybrid filler were characterised by a loose structure, and their distribution was fairly homogeneous.By using PVAc as an adhesive, it was possible to achieve partial distribution of carbon nanotubes on the CF’s surface. In addition, its presence can contribute to an increase in the filler–matrix adhesive influence.PVC composites with a carbon hybrid filler are characterised by a substantially higher rigidity and comparable impact strength when compared to PVC. No improved tensile strength was identified, which was due to the filler’s anisotropic distribution, uneven adhesion, and wettability, but also due to the presence of carbon aggregates which acted as a discontinuity of the composite structure.The thermal stability analysis allowed for the conclusion that the developed filler did not shorten the thermal stability, but it slightly increased its temperature (T5).A slight increase in the T_g_ of the composites’ matrix observed at the maximum share of hybrid filler in the matrix was related to the increased share of nanofiller and its interaction with the matrix’s macromolecules.The hybrid filler substantially changed the materials’ electrical properties, thereby lowering the volume resistivity. The proposed method of hybrid filler production did not cause structural defects in the filler’s components, but it increased the quality of nanoparticle dispersion, thereby enabling the creation of a conductive, interpenetrating network on micro and nanometric scales.Further research will be focused on developing a method of modifying the carbon hybrid filler to improve its mechanical properties and to maintain its electrical properties.

## Figures and Tables

**Figure 1 materials-15-05625-f001:**
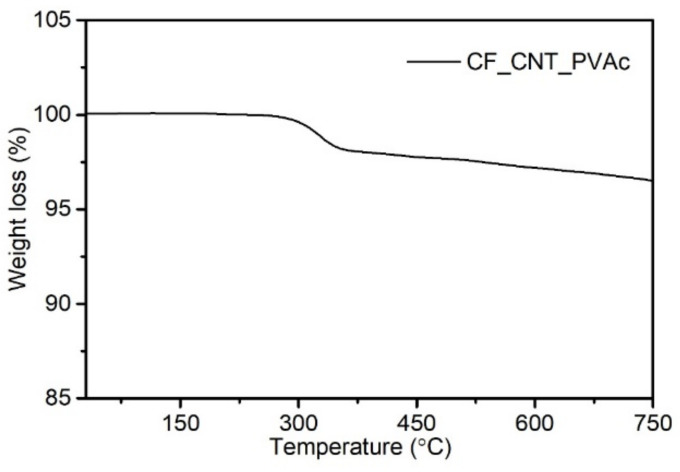
TG plot of carbon hybrid filler CF/CNT-PVAc.

**Figure 2 materials-15-05625-f002:**
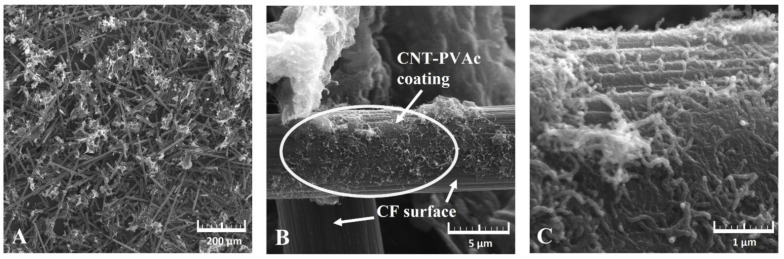
SEM images of carbon hybrid filler CF/CNT-PVAc ((**A**)—mag. 200×; (**B**)—mag. 10 k×; (**C**)—mag. 50 k×).

**Figure 3 materials-15-05625-f003:**
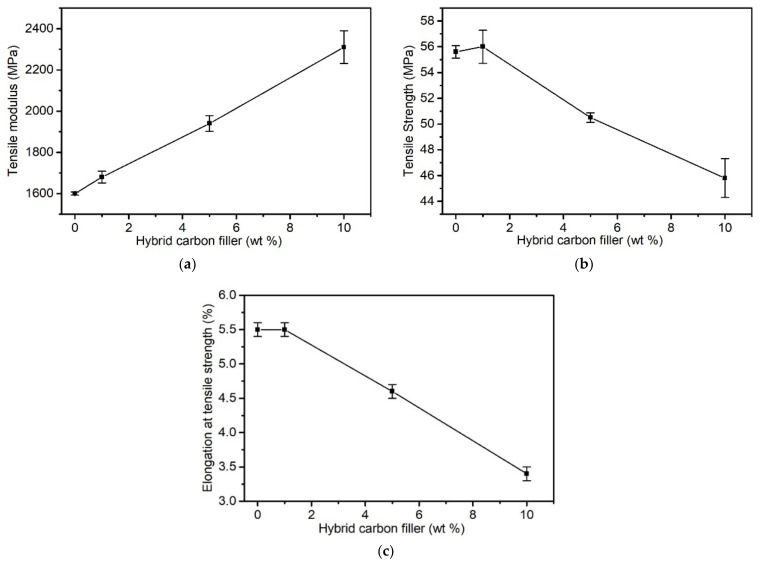
Mechanical properties of the PVC CF/CNT–PVAc composites as a function of hybrid filler concentration (**a**) Tensile modulus (**b**) Tensile strength (**c**) Elongation at tensile strength.

**Figure 4 materials-15-05625-f004:**
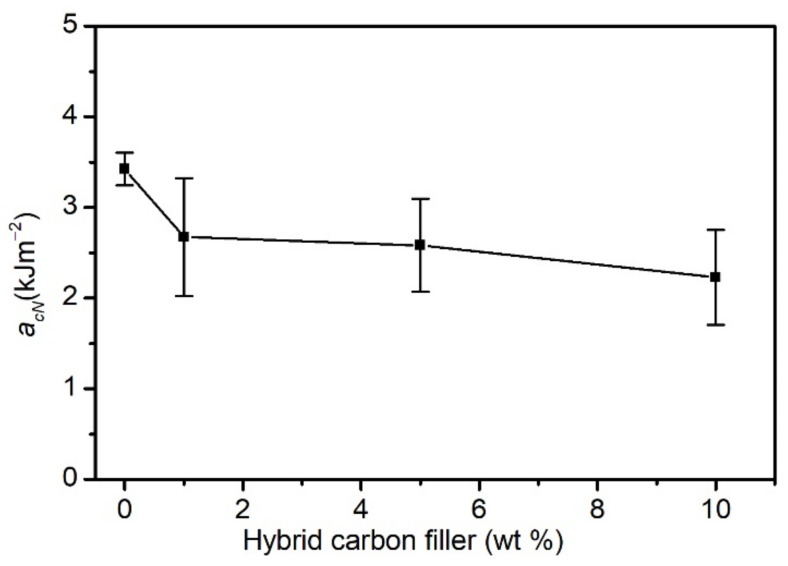
Impact strength of PVC CF/CNT–PVAc composites.

**Figure 5 materials-15-05625-f005:**
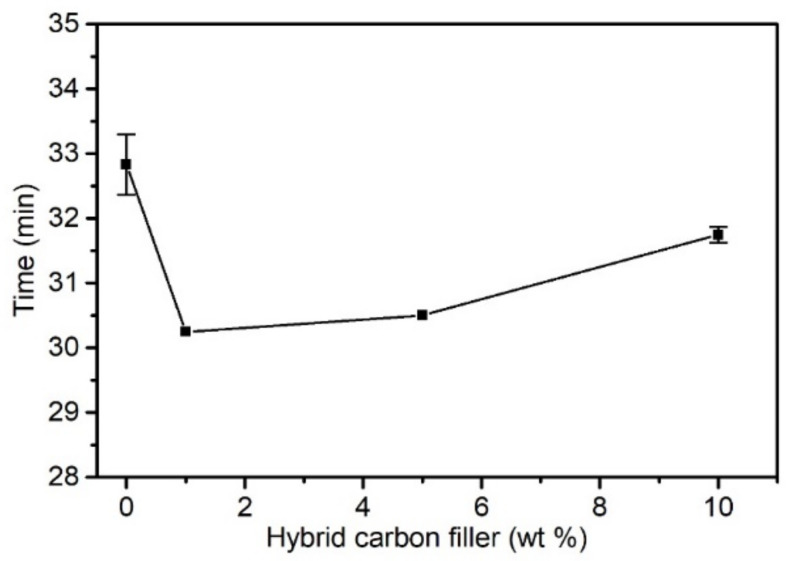
Time of thermal stability as a function of hybrid filler concentration in PVC matrix.

**Figure 6 materials-15-05625-f006:**
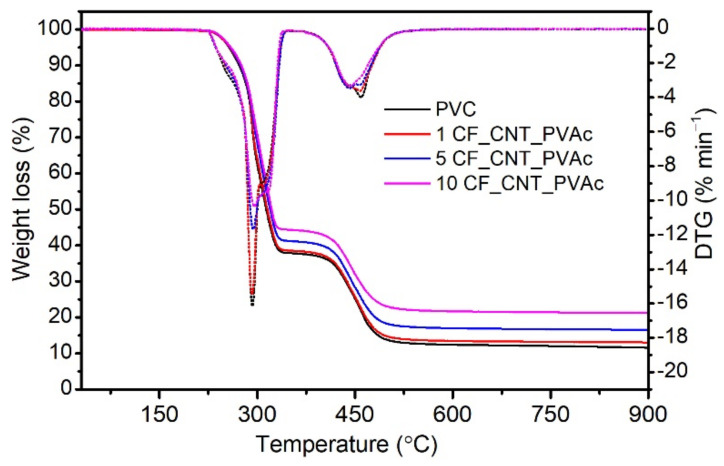
TG and DTG thermograms of PVC/hybrid filler composites tested in a nitrogen atmosphere.

**Figure 7 materials-15-05625-f007:**
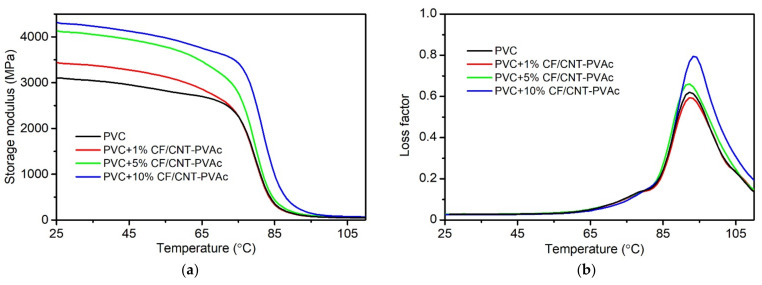
Storage modulus and tan δ of PVC and PVC hybrid filler composites. (**a**) Storage modulus (**b**) loss factor.

**Figure 8 materials-15-05625-f008:**
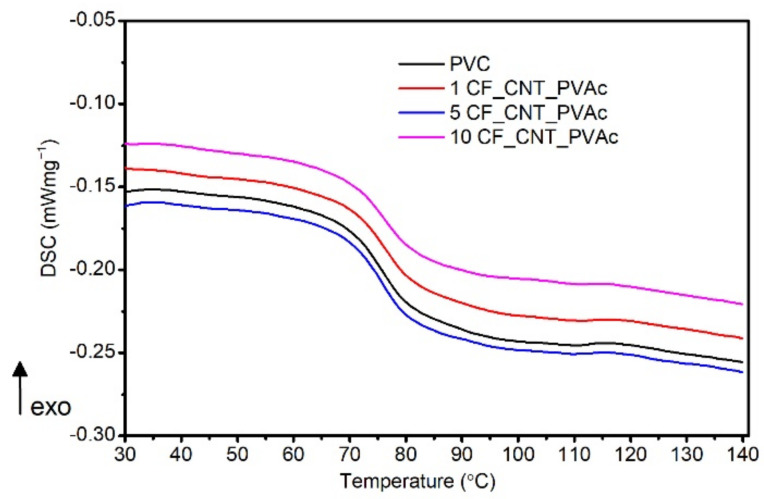
Glass transition region of PVC materials investigated by DSC.

**Figure 9 materials-15-05625-f009:**
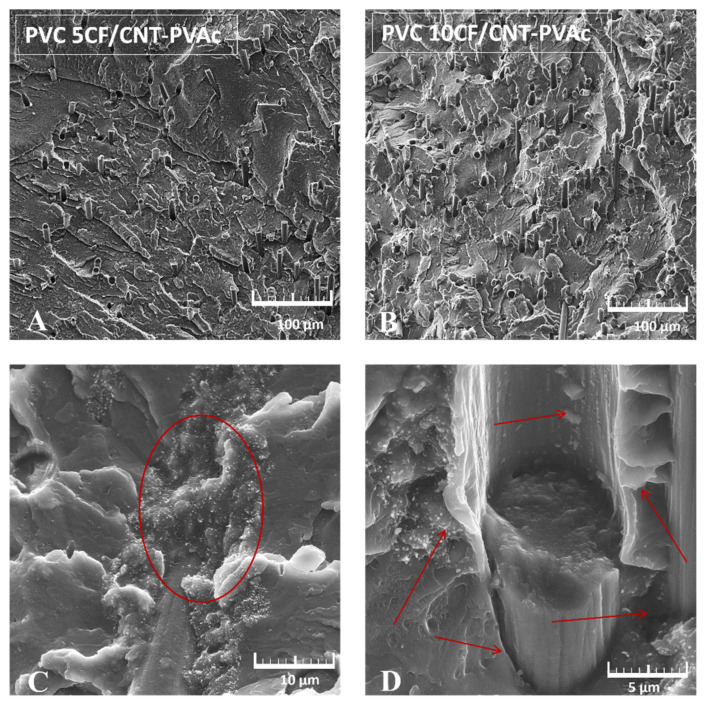
SEM observations of cryogenic fracture of PVC hybrid filler composites. (**A**) PVC 5CF/CNT-PVAc - mag. 500× (**B**) PVC 10CF/CNT-PVAc - mag. 500× (**C**) fibre–matrix interface - mag. 5 k× (**D**) fibre–matrix interface mag. 10 k×.

**Figure 10 materials-15-05625-f010:**
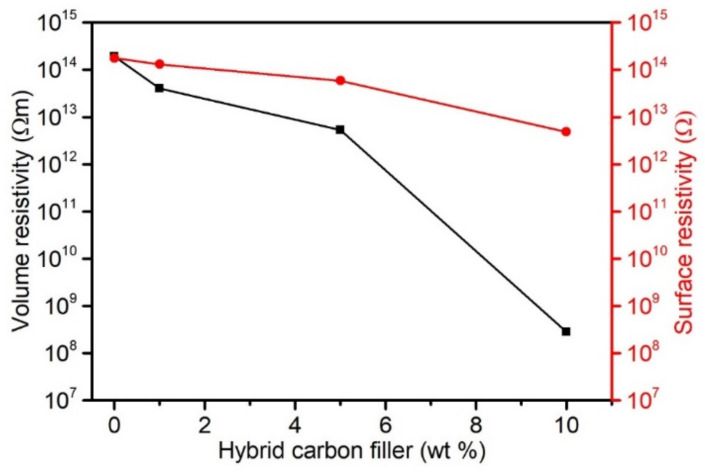
Volume and surface resistivity of PVC materials as a function of hybrid filler content.

**Figure 11 materials-15-05625-f011:**
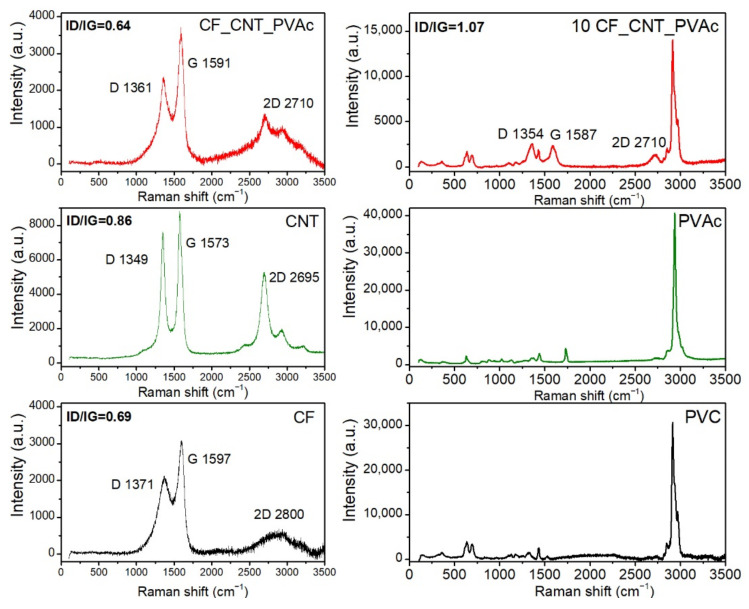
Raman Investigation of CF, CNT, hybrid filler, PVC, PVAc, and PVC composites modified by 10 wt% of hybrid filler.

**Table 1 materials-15-05625-t001:** The TG analysis results.

Filler Content (wt%)	T_5_ (°C)	T_10_ (°C)	T_50_ (°C)	T_DTG_ (°C)	RM at 900 °C (%)
0	254.5	270.4	313.8	291.5	11.3
1	258.1	274.3	315.2	291.1	13.2
5	259.2	274.5	319.8	292.5	16.6
10	260.1	275.8	321.8	292.3	21.3

**Table 2 materials-15-05625-t002:** Storage modulus at different temperatures and values of T_g_ of PVC and its composites.

Filler Content (wt%)	E′ 25 °C (MPa)	E′ 40 °C (MPa)	E′ 60 °C (MPa)	E′ 80 °C (MPa)	E′ 90 °C (MPa)	T_g E′_ (°C)	T_g tanδ_ (°C)
0	3115.0	3008.5	2769.4	1268.8	144.4	74.9	92.3
1	3427.1	3335.0	3017.6	1232.3	138.1	73.6	92.6
5	4117.6	4008.0	3658.1	1512.8	179.0	73.7	92.0
10	4296.4	4186.6	3878.2	2517.0	334.1	76.6	93.6

**Table 3 materials-15-05625-t003:** The T_g_ results of investigated materials.

Filler Content (wt%)	T_g Mid_ (°C)	T_g Infl_ (°C)
0	75.7	75.4
1	75.9	75.3
5	75.8	74.5
10	76.2	77.4

## Data Availability

The data that support the findings of this study are available from the corresponding author (Katarzyna Skórczewska) on request.

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
