# Peer review of "Novel Composites of Poly(vinyl chloride) with Carbon Fibre/Carbon Nanotube Hybrid Filler"

_materials, 2022, doi:10.3390/ma15165625_

Round 1

Reviewer 1 Report

The hybrid filler (CF/CNT) which prepared by the solution method was blending with poly(vinyl chloride) (PVC) to improve the performance of PVC. This research is of great significance for modifying the carbon hybrid filler to improve the performance of materials. I think the manuscript need to be minor revise before publish.

(1) In the introduction part, the “dispersion” in the second paragraph has no clear correlation with the “wettability” in the third paragraph.

(2) Line “167-170” DMA ”and “DSC” need to be defined.

(3) The dimensions of impact splines need to be indicated in “2.4.2”.

(4) Line “523-525” "In the case of" has a syntax error.

(5) Line 524 what is mean of “PP-CF-CNT”.

(6) The "1.354" in line 576 is incorrectly stated.

(7) The reference format needs to be carefully checked, such as the reference [104],[105], the year is not bolded and the journal is not italicized.

Author Response

Dear Reviewer, 

Thank you for your valuable comments to the submitted manuscript.  

We would like to respond to your comments. 

(1) In the introduction part, the “dispersion” in the second paragraph has no clear correlation with the “wettability” in the third paragraph

The mentioned paragraphs deal with two different fillers: nanometric carbon structures and micrometric carbon fibers. The first one describes the problem of dispersibility of nanoparticles in the polymer matrix, which is the main problem to overcome during the manufacture of polymer nanocomposites. Their strong tendency to agglomerate is a result of their nanometric dimensions.

In contrast, in the case of CFs, due to their micrometric dimensions, achieving adequate dispersion is not problematic. However, the issue of CF-polymer wettability is a key problem. A number of scientific papers have been devoted to this issue. They describe methods of increasing this wettability. One such method is the modification of the CF surface, including applying carbon coatings in the form of carbon nanotubes, as we mentioned in the literature section.

(2) Line “167-170” “DMA ”and “DSC” need to be defined.

We thank you for your comment, appropriate corrections have been made in the manuscript.

(3) The dimensions of impact splines need to be indicated in “2.4.2”.

Thank you for your comment.

Rectangular specimens with a length of 80 mm were used. A dimension of a cross-sectional at the bottom of notch was about 8 mm x 4 mm. A type A notch according to ISO 179-1 was employed.

Appropriate changes have been made in the manuscript.

(4) Line “523-525” "In the case of" has a syntax error.

Thank you for the valuable suggestion. Part of the sentence was missing. Correctly is: "The authors suggest, based on SEM observations, that MWCNTs do not locate flat on the CF surface, but are spatially localized in the matrix, which contributes to the formation of spatial conduction paths between CF and MWCNTs." We have made the appropriate corrections in the manuscript.

(5) Line 524 what is mean of “PP-CF-CNT”.

We thank you for your comment. The acronym used has been expanded: Polypropylene/ carbon fibres/carbon nanotubes hybrid composites. Appropriate corrections have been made in the manuscript.

(6) The "1.354" in line 576 is incorrectly stated.

We thank you for your comment, appropriate corrections have been made in the manuscript.

(7) The reference format needs to be carefully checked, such as the reference [104],[105], the year is not bolded and the journal is not italicized.

We thank you for your comment, appropriate corrections have been made in the manuscript.

Reviewer 2 Report

This study investigated the characterization and mechanical properties of PVC/hybrid filler composites. Polyvinyl chloride, carbon fibres, multiwalled carbon nanotubes and hybrid combination were used to examine the composite. Although the article is written well, it requires some revisions before it can be accepted for publication.

1. Abstract: The description of the work investigated should be discussed more in detail. Include the numbers of mixtures, the dosage of fibres, parameters investigated and methods used.

2. Literature sections are clustered with the group of citations. It is suggested to add a few and discuss individual literature.

3. It is suggested to include the physical properties of PVC and MWCNT.

4. On what basis is the percentage (1, 5 and 10) of PVC composites selected?

5. Lines 186, 192, cite the standards. The same is applicable for other standards wherever applicable.

6. Lines 191-195, Mention the size of sample used for the impact test.

7. Figures 1,3,4,5,7,8, 10, 11 are not clear and blurred. Please improve the clarity of Figures.

8. Why are the volume resistivity and surface resistivity higher for 10% hybrid filler content?

9. The discussion sections are discussed adequately with the supporting literature.

10. The conclusions are clear and reflect the key findings.

Author Response

Dear Reviewer, 

Thank you for your valuable comments on the submitted manuscript.  

We would like to respond to your comments. 

1. Abstract: The description of the work investigated should be discussed more in detail. Include the numbers of mixtures, the dosage of fibres, parameters investigated and methods used.

Thank you for your valuable comment. The abstract has been improved following the Reviewer's instructions. Appropriate changes have been made in the manuscript.

2. Literature sections are clustered with the group of citations. It is suggested to add a few and discuss individual literature.

Thank you for your suggestion. However, we would like to leave this part of the manuscript without the suggested changes.

Groups of citations contain similar information. Developing them would make the entire manuscript excessively long. A lot of the literature has been expanded individually in discuss of the results.

3. It is suggested to include the physical properties of PVC and MWCNT.

Thank you for your suggestion. Appropriate changes as suggested by the Reviewer have been made in the text.

4. On what basis is the percentage (1, 5 and 10) of PVC composites selected?

Polymer composites with nanofillers are generally produced with a small amount of nanofiller, unlike carbon fiber composites. In our case, the introduction of 1% hybrid filler into PVC corresponds to a share of nanotubes of about 0.0005wt%.

We used a PVC compound with a poor composition. This was to reduce the adulteration of the interactions between filler and PVC. Dry blend didn’t contain modifiers necessary for proper processing of PVC with a high proportion of filler. A filler content larger than 10% would significantly increase the viscosity of the processed composite, thereby causing an increase in shear forces. As a results, it may degrade not only a filler but also the matrix.

5. Lines 186, 192, cite the standards. The same is applicable for other standards wherever applicable.

We thank the Reviewer for his comment, appropriate modifications have been made in the manuscript

6. Lines 191-195, Mention the size of sample used for the impact test.

We thank the Reviewer for his comment, appropriate modifications have been made to the manuscript.

Rectangular specimens with a length of 80 mm were used. A dimension of a cross-sectional at the bottom of notch was about 8 mm x 4 mm. A type A notch according to ISO 179-1 was employed.

7. Figures 1,3,4,5,7,8, 10, 11 are not clear and blurred. Please improve the clarity of Figures.

The blurring is probably due to the transformation of the manuscript from doc to pdf format, which is also evident in the case of the journal logo on the first page of the manuscript.

Please find attached the figures we have entered. We will report the problem to the Editor. 

8. Why are the volume resistivity and surface resistivity higher for 10% hybrid filler content?

We thank the Reviewer for this question.

According to our observations, PVC composites with 10wt% hybrid filler reached the lowest value of the volume and surface resistivity. This is due to the formation of a spatial interpenetrating conduction network built of micrometric and nanometric carbon structures. With an increase of the hybrid filler content in the PVC matrix, the proportion of effectively conducting carbon nanotubes increases, and the density of the formed spatial conduction network also increases. The CNT dispersed in the form of sheets/bucky paper fragments and CNT dispersion on CF surface improved the contact density, which ultimately resulted in more effective charge conduction at smaller nanofiller content according to the conduction mechanism via the CNT structure and direct contact or according to quantum tunnelling. An explanation of this phenomenon is provided in the manuscript in verses 547-557.

9. The discussion sections are discussed adequately with the supporting literature.

We thank the Reviewer for the comment.

10. The conclusions are clear and reflect the key findings.

We thank the Reviewer for the comment.

Reviewer 3 Report

In the manuscript “Novel Composites of Poly(vinyl chloride) with Carbon Fibres/Carbon Nanotubes Hybrid Filler”, the authors present comprehensive study of the CF-CNTs-PVAc hybrid filler, including the fabrication, the mechanical properties, thermal stabilities, electrical conductivities, and so on. Overall, the developed filler demonstrated higher rigidity and comparable impact strength, comparable thermal stability, lower volume resistivity when compared to PVC. I think this manuscript is well written and can be published in Material after minor revisions. I also have a few questions/comments to authors for further clarification and make some remarks regarding the manuscript and data presentation.

1.     For the SEM images of the carbon hybrid filler, the authors should clearly label the different compositions of carbon fibers, CNTs, and PVAs. The authors describe the SEM image as: “As indicated by the filler’s SEM images (Figure 2), the CF surface is only coated with CNT or the adhesive compound (PVAc) in some sections, but the majority is maintained in the form of original fibre”. It’s a bit confusing for readers, about which part of the hybrid is the authors referring to?

2.     The Raman results and discussion (Fig. 11) can be moved to section 2, which is more related to the characterization of the hybrid filler instead of test results.

Author Response

Dear Reviewer, 

Thank you for your valuable comments to the submitted manuscript.  

We would like to respond to your comments. 

  1. For the SEM images of the carbon hybrid filler, the authors should clearly label the different compositions of carbon fibers, CNTs, and PVAs. The authors describe the SEM image as: “As indicated by the filler’s SEM images (Figure 2), the CF surface is only coated with CNT or the adhesive compound (PVAc) in some sections, but the majority is maintained in the form of original fibre”. It’s a bit confusing for readers, about which part of the hybrid is the authors referring to?

We thank the Reviewer for the important suggestion. The different components of the obtained hybrid filler have been labeled on the SEM images (fig.2), in such a way as to make the perception more accessible and clear to the reader.

  1. The Raman results and discussion (Fig. 11) can be moved to section 2, which is more related to the characterization of the hybrid filler instead of test results.

We thank the Reviewer for this comment, however, we would like to leave the hybrid filler description here. In the description of the composite, we would largely have to duplicate the information presented in the description of the CF, CNT, and hybrid filler. Therefore, leaving these descriptions in one section seems justified to us.